# Innovative Antimicrobial Chitosan/ZnO/Ag NPs/Citronella Essential Oil Nanocomposite—Potential Coating for Grapes

**DOI:** 10.3390/foods9121801

**Published:** 2020-12-04

**Authors:** Ludmila Motelica, Denisa Ficai, Anton Ficai, Roxana-Doina Truşcă, Cornelia-Ioana Ilie, Ovidiu-Cristian Oprea, Ecaterina Andronescu

**Affiliations:** 1Faculty of Applied Chemistry and Material Science, University POLITEHNICA of Bucharest, Spl. Independentei 313, 060042 Bucharest, Romania; motelica_ludmila@yahoo.com (L.M.); denisa.ficai@upb.ro (D.F.); anton.ficai@upb.ro (A.F.); roxana.trusca@upb.ro (R.-D.T.); cornelia.ilie18@gmail.com (C.-I.I.); ecaterina.andronescu@upb.ro (E.A.); 2Section of Chemical Sciences, Academy of Romanian Scientists, Ilfov st. 3, 050045 Bucharest, Romania

**Keywords:** antimicrobial packaging, chitosan, ZnO, Ag nanoparticles, citronella essential oil

## Abstract

New packaging materials based on biopolymers are gaining increasing attention due to many advantages like biodegradability or existence of renewable sources. Grouping more antimicrobials agents in the same packaging can create a synergic effect, resulting in either a better antimicrobial activity against a wider spectrum of spoilage agents or a lower required quantity of antimicrobials. In the present work, we obtained a biodegradable antimicrobial film that can be used as packaging material for food. Films based on chitosan as biodegradable polymer, with ZnO and Ag nanoparticles as filler/antimicrobial agents were fabricated by a casting method. The nanoparticles were loaded with citronella essential oil (CEO) in order to enhance the antimicrobial activity of the nanocomposite films. The tests made on Gram-positive, Gram-negative, and fungal strains indicated a broad-spectrum antimicrobial activity, with inhibition diameters of over 30 mm for bacterial strains and over 20 mm for fungal strains. The synergic effect was evidenced by comparing the antimicrobial results with chitosan/ZnO/CEO or chitosan/Ag/CEO simple films. According to the literature and our preliminary studies, these formulations are suitable as coating for fruits. The obtained nanocomposite films presented lower water vapor permeability values when compared with the chitosan control film. The samples were characterized by SEM, fluorescence and UV-Vis spectroscopy, FTIR spectroscopy and microscopy, and thermal analysis.

## 1. Introduction

Because of alteration or spoilage generated by microorganisms or simply due to shelf life expiration, about 40–50% of fruits and vegetables, 35% of fish, 30% of cereals, and 20% of dairy and meat products are lost yearly. In less developed countries, the largest share of losses occurs after harvesting, until the processing phase, while in developed countries, the largest contribution to food waste is at the retail and customer level [1]. Most food packaging used at present is made from cellulosic materials or petrochemical polymers due to historic factors like good barrier performances or low-cost [2]. While cellulose-based materials are biodegradable [3], the plastics are not and worldwide, they have already raised a lot of environmental concerns regarding short and long term pollution [4]. Nevertheless, neither of these two materials present antimicrobial activity.

These factors are pressuring the food industry to develop new types of antimicrobial, biodegradable packaging materials. Such innovative materials can actively control bacterial and fungal proliferation, diminishing food spoilage, enhancing food quality and safety, and ensuring a longer shelf life. While many biopolymers like cellulose, starch, alginate, etc. are tested as packaging materials, only few have antibacterial activity [5]. Chitosan is one of the most abundant biopolymers and is the most studied for edible packaging materials, but the simple chitosan films have a low mechanical resistance and poor barrier properties [6,7,8]. Chitosan is a polysaccharide with some intrinsic antibacterial activity, especially against Gram-negative strains [9], which in most cases, is extracted from crab or shrimp shells, but can also be sourced from mushrooms inferior stem [10]. There are many proposed pathways that can explain the chitosan antibacterial activity [11] and thus, how it can protect food. In order to improve its mechanical properties, some plasticizers are added and chitosan has very good compatibility with glycerol [12]. Adding nanoparticles to the biopolymeric film will also reinforce it, enhancing the mechanical and barrier properties. Some nanoparticles like ZnO can be embedded directly into the chitosan matrix, acting like crosslinking agents, but for others, like AgNPs (silver nanoparticles), a surface modification is advisable [13].

The antibacterial activity of ZnO has been known for some time, but its antimycotic activity is seldom investigated [14]. Its exact antimicrobial mechanisms are not known, but there are at least two separate pathways with respect to the light presence. The first mechanism implies photocatalytic activity and perhaps reactive oxygen species (ROS) production, which are responsible for oxidative stress, which damages the bacterial membrane. The second mechanism does not require light and is based on nanoparticle internalization and mechanical damage (like puncture and rupture) to the microorganism membrane [15].

Silver nanoparticles (AgNPs) are well known antimicrobial agents, with their activity being documented for over 650 microorganisms (bacteria, fungi, or even viruses). The antimicrobial activity is influenced by size and shape of the AgNPs, with smaller ones being more potent [16].

Plant extracts and essential oils are thoroughly investigated for their huge potential as antimicrobial and antioxidant agents [17,18,19]. Citronella essential oil (CEO) contains, among other substances, geraniol, citronellal, and citronellol as its major constituents [20]. It is considered a biopesticide with a non-toxic mode of action in the US [21]. It also has a strong antifungal activity [22]. Chitosan films have poor performance regarding water vapor permeability (WVP), but this drawback can be removed by adding plant extracts or essential oils into its composition as these increase its hydrophobicity [23,24,25].

Chitosan is one of the most studied solutions for edible films that can replace the waxy coatings of fruits and confer antimicrobial activity together with a longer shelf life [11]. Chitosan has been found to be non-toxic and is approved by the United States Food and Drug Administration (US-FDA) as a generally recognized as safe (GRAS) material [26]. Zinc oxide is also considered GRAS by the US-FDA and due to its good antimicrobial properties, is suitable to being applied as an active compound in food packaging [27,28,29]. AgNPs are one of the most potent antimicrobials with great potential regarding applications in the food industry [30]. The use of nanomaterials in food coatings has some major advantages, such as strong antimicrobial activity and a UV-barrier, which will extend the shelf life, but there is a constant need to evaluate the risks associated with metallic and oxide NPs. The NPs interaction with microbial cells is the foundation of their antimicrobial activity, but at the same time, they can interact with human cells. The use of a mixture of two types of nanoparticles (Ag and ZnO) and a natural antimicrobial agent (CEO) permits due to the synergic action of the lowering of NPs concentration. While ZnO upon reaching the stomach can be assimilated as a beneficial mineral source, further studies still need to be carried out in order to assess its safety toward the consumers’ exposure [31]. AgNPs associated toxicity can be decreased by using a very low concentration [32,33,34]. For CEO, the studies report no toxicity at the used concentration, but there are some concerns about the impact on beneficial gut bacteria [35,36].

In the literature, packaging materials with composition chitosan–ZnO [31] or chitosan–AgNPs [37] are reported and there are some studies with antimicrobial activity of ZnO/AgNPs composites. Previous reports of mixing ZnO and AgNPs indicate an enhancing of antimicrobial performance [38,39], especially against *Staphylococcus aureus*, and to a lesser extent, against *Escherichia coli*. In both studies, ZnO NPs concentration was fixed and only Ag NPs varied. In experiments where the support’s surface loading was not controlled, the results did not indicate an enhancing antibacterial activity, probably due to different NPs concentrations [40]. If different capping agents were used for NPs, the antibacterial activity was not influenced significantly, but the antifungal activity against *Candida albicans* was better for NPs capped with 4-styrenesulfonic acid and maleic acid [41].

The objective of this research was to obtain a biodegradable antimicrobial film that can be used as packaging material for extending the shelf life of fresh fruits. Therefore, chitosan was chosen as a base biopolymer, with ZnO and Ag nanoparticles as filler/antimicrobial agents for the fabrication of nanocomposite films. The nanoparticles were loaded with citronella essential oil in order to enhance the antimicrobial activity. To the best of our knowledge, this is the first time a chitosan-based film with both types of nanoparticles, ZnO and Ag, loaded with citronella essential oil has been obtained. Moreover, the innovative films were also compared with the simple chitosan–ZnO–CEO and chitosan–AgNPs–CEO to study the potential influences and synergisms. The biological tests indicate that, indeed, a synergic effect was obtained, with broad-spectrum activity as all components had antimicrobial activity.

## 2. Materials and Methods

### 2.1. Materials

Zinc acetate dihydrate with 99.9% purity, absolute ethanol, and silver nitrate were obtained from Merck. Chitosan (CAS 9012-76-4), glycerol, acetic acid, polyvinylpyrrolidone (PVP), Sabouraud dextrose broth, Nutrient Broth, and agar were obtained from Sigma Aldrich (Redox Lab Supplies, Bucharest, Romania). Citronella essential oil (CEO) was purchased from Carl Roth (Amex-Lab, Bucharest, Romania).

The chemicals were used without any further purification.

### 2.2. Synthesis of ZnO and Ag Nanoparticles

ZnO synthesis was done as described in [42]. Briefly, 2.1950 g zinc acetate dihydrate was solved in 50 mL absolute ethanol and heated under magnetic stirring at boiling point. After 10 h, the precipitate was washed and centrifugated three times and the resulting powder was dried at 105 °C.

AgNPs were synthetized as described in [43]. Then, 0.3 g AgNO_3_ was dissolved in 100 mL water under vigorous stirring at 70 °C. Then, 10 mL solution of 0.1 g trisodium citrate was added dropwise as a reduction agent. After 30 min, a third solution (0.1 g PVP in 10 mL) was added dropwise. The yellow solution containing AgNPs was used without further purifications.

### 2.3. Synthesis of Chitosan/ZnO and/or AgNPs Films

A ZnO suspension in 15 mL water mixed with 1 mL CEO and the solution was sonicated for 30 min before being used (for CZ and CZA films, Table 1). Then, 1 mL CEO was added to various volumes of AgNPs solution and the mix was sonicated for 30 min before being used (for CA and CZA films).

Chitosan blends were prepared using the solvent casting methods. Then, 0.66 g chitosan was dissolved in 100 mL of 1% (*v/v*) acetic acid solution and 2 mL of glycerol was added after complete solubilization of the chitosan. To this solution, under vigorous stirring, the above obtained ZnO and/or AgNPs suspensions were added to obtain the ratios presented in Table 1.

Each solution was casted in a Petri dish and all were dried in an oven for 48 h at 40 °C. A control film without nanoparticles or CEO was prepared similarly. After removal from the Petri-dish, the films were stored in ziplock plastic bags at 20 °C and 60% relative humidity.

The concentration of ZnO and AgNPs for the synthesis of CZA films was specially chosen to obtain two series with variable amounts of ZnO or AgNPs. The use of the mixture of two nanoparticles and a natural antimicrobial agent was chosen based on the approaches reported in the literature, especially the synergic and potentiating action [44,45]. Therefore, in the series CZA1, CZA2, and CZA3, the quantity of AgNPs remains constant, while the ZnO NPs concentration increased. For the series CZA4, CZA2, and CZA5, the amount of ZnO NPs remained constant and the concentration of AgNPs increased. The same quantity of CEO (1 mL) was added to each film.

### 2.4. Characterization of Chitosan Composite Films

#### 2.4.1. Microstructural Analysis

Scanning electron micrographs for determination of the films surface morphology and microstructure were obtained using a QUANTA INSPECT F50, FEI Company, Eindhoven, The Netherlands scanning electron microscope equipped with field emission gun—FEG with 1.2 nm resolution and an energy dispersive X-ray spectrometer (EDS) with an MnK resolution of 133 eV.

#### 2.4.2. Fourier Transform Infrared Spectroscopy

The presence of functional groups and interactions between various components of the composite films were investigated by using Fourier transform infrared spectroscopy (FTIR) in the wavenumber range 4000–400 cm^−1^. The spectra were recorded with a Nicolet iS50 FTIR spectrometer (Nicolet City, MA, USA), equipped with a DTGS detector, at a resolution of 4 cm^−1^, by averaging 32 scans.

In order to obtain information about spatial distribution of the components, FTIR 2D maps were recorded with a FTIR microscope Nicolet iS50R (Nicolet City, MA, USA), with DTGS detector, in the wavenumber range 4000–600 cm^−1^.

#### 2.4.3. Photoluminescence Spectroscopy

Photoluminescence spectrum (PL) was measured with a Perkin Elmer (Waltham, MA, USA) P55 spectrometer using a Xe lamp as a UV light source at ambient temperature, in the range 350–600 nm. The measurement was made with a scan speed of 200 nm min^−1^, excitation and emission slits of 10 nm, and a cut-off filter of 1%. An excitation wavelength of 320 nm was used.

#### 2.4.4. UV-Vis Spectroscopy

Diffuse reflectance spectra measurements were made with a JASCO (Easton, PA, USA) V560 spectrophotometer with solid sample accessory, in the domain 200–900 nm, with a speed of 200 nm min^−1^.

#### 2.4.5. Thermal Analysis

For thermal analysis, TG-DSC, approximatively 10 mg from each sample, was placed in an open alumina crucible in a Netzsch (Selb, Germany) STA 449C Jupiter apparatus. The heating was done from room temperature up to 900 °C, with 10 K∙min^−1^, under a flow of 50 mL∙min^−1^ dried air. An empty alumina crucible was used as a reference.

#### 2.4.6. Water Vapor Permeability (WVP)

The WVP was determined as described in [46] by using permeation cups of 50 mm diameter sealed with a sample film. Each cup contained 10 g of dried CaCl_2_. The permeation cups were put in a box at a temperature of 25 °C and 75% relative humidity. Their weight was measured at fix intervals (8 h) for four days.

### 2.5. Antimicrobial Assay

The samples were placed in Petri dishes, with specimens of (6 mm × 6 mm). To avoid the impact of contaminants on the experiment, all specimens used were previously sterilized as described in [3].

The antimicrobial activity was evaluated against *Candida albicans* ATCC 10231; *Staphylococcus aureus* ATCC 25923; and *Escherichia coli* ATCC 25922. Microbial cell suspensions were made in physiological buffer from fresh cultures (24–48 h) with a standard density of 1.5 × 10^8^ CFU/mL (corresponding 0.5 McFarland standard) for *S. aureus* and *E. coli* and 3 × 10^8^ CFU/mL (corresponding 1 McFarland standard) for *C. albicans*, developed on Nutrient Broth with 2% agar for bacterial strains (NBA) and Sabouraud dextrose broth with 2% agar for yeast (SA).

For qualitative antimicrobial assay used the diffusometric method and the tested strains were inoculated by insemination in the “canvas” with a tampon on the surface of solid agar medium, NBA, and SA in Petri dishes. Samples of the same size were added at equal distances. For controls, solutions (CEO, Ag NPs and from a 100 mg/mL ZnO suspension) drops with a volume of 10 μL were added as spots at equal distances. The Petri dishes were incubated for 24 h for *S. aureus* and *E. coli* and 48 h for *C. albicans* at 37 °C and after that, the sensibility of microbial strains to all compounds from materials/nanostructured films that were diffused on the medium surface was assessed by measuring the inhibition diameters. All experiments were designed and performed in triplicates.

### 2.6. Statistical Analysis

The data from the antimicrobial assay was analyzed using the analysis of variance (ANOVA) with the help of Microsoft Excel 2016 (Microsoft Corp., Redmond, WA, USA), with XLSTAT 2020.5.1 add-on. Normal distribution of the groups was checked with the Shapiro-Wilk test; homoscedasticity of the residuals was assessed by Levene’s test; and Tukey’s (HSD) test was used to compare the results and reveal the pairs of films that differed in terms of statistical significance, where *p* < 0.05.

## 3. Results and Discussion

### 3.1. Chitosan Films Characterization

The films were grouped in lots—the chitosan/ZnO/CEO with samples coded from CZ1 to CZ4 plus a control chitosan sample, noted C; the chitosan/AgNPs/CEO with samples coded from CA1 to CA4; and the chitosan/ZnO&AgNPs/CEO with codes CZA1 to CZA5, as indicated in Table 1. Similar samples from all films were analyzed by means of SEM, FTIR, UV-Vis, PL, and TG/DSC.

### 3.2. FTIR Spectroscopy and Microscopy

#### 3.2.1. FTIR Spectroscopy

The FTIR spectra was used to identify the modifications induced by the presence of ZnO, Ag NPs, and CEO to the polymeric matrix. The main absorption peaks and their assignment are presented in Table 2. A strong, broad band in the region 3100–3500 cm^−1^ corresponds to O-H and N-H vibrations in the chitosan control film [47]. The shift in the presence of ZnO indicates the interactions between polymeric chains and nanoparticles surface, mediated by these moieties [48]. The peak from 2921 cm^−1^ remains largely unchanged as this is attributed to C-H symmetric vibration. Interactions between residual N-acetyl groups and nanoparticles are also indicated by the shifts of amide I band from 1651 to 1643 cm^−1^. These shifts are also reported in the literature for chitosan-ZnO interactions [49,50]. The presence of ZnO nanoparticles is also indicated by the intense absorption peaks in the 400–500 cm^−1^ region [51].

The FTIR spectra can also be used, to a lesser extent, to identify the modifications induced by the presence of AgNPs and CEO to the chitosan structure. The shifts to the band corresponding to O-H and N-H vibrations are less important, indicating only weak interactions of chitosan with AgNPs. Also, the peaks from 2921 cm^−1^ remain unchanged. Some weak interactions between residual N-acetyl groups and AgNPs are also indicated by the small shifts of amide I band from 1651 to 1647–1644 cm^−1^. The changes in intensity for the OH, CO, and NH characteristic vibration bands can also be considered as an indication of which chitosan groups are contributing to the interactions [52].

#### 3.2.2. FTIR Microscopy

The spatial distribution of ZnO, Ag NPs, CEO, and chitosan was also investigated by FTIR microscopy.

In Figure 1, the FTIR maps recorded at 3275, 1743, and 1643 cm^−1^ for the chitosan control and CZ1–CZ4 films are presented. The distribution of CEO and ZnO nanoparticles on the surface of the chitosan films is quite homogenous, with occasional small agglomerated clusters/accumulations or defects. The membranes C, CZ1–CZ3 are quite homogeneous as this can be proven based on the two maps recorded at 3275 and 1643 cm^−1^ and only some minor differences are presented, with these changes being at most, tens of µm. In the case of CZ4, there are some noticeable differences between the maps recorded at 3275, 1743, and 1643 cm^−1^, which means that more important heterogeneities are present, especially because of a more heterogeneous distribution of the CEO and ZnO NPs within the chitosan film. Generally, it can be concluded that these membranes are more uniform than the samples CA1–CA4 because of the presence of larger amounts of ZnO NPs, which interact well with the CEO and do not allow the formation of the emulsions.

In Figure 2, the FTIR maps recorded for CA1–CA4 films at ~1740 and 3500 cm^−1^, representing a specific band of CEO and the maximum of the hydrogen bonds are presented. Based on the two series, one can clearly conclude that the distribution of the bands is quite different, which means that during the formation of the membrane, the CEO and chitosan gel formed a water/oil emulsion, which remained even after drying and this morphology can be clearly visualized in the video image of CA4 (in fact, all these series have the same morphology) [53].

For the CZA1–CZA5 films, the spatial distribution of ZnO and AgNPs was also investigated by FTIR microscopy, Figure 3. The recorded map images are presented for some representative wavenumbers at 3293 and 1643 cm^−1^. Based on the compositions of the CZA series as well as the compositions of the CZ series, it is expected to obtain a similar behavior. Indeed, also in this case, a good distribution of the components can be observed, with ZnO adsorbing the CEO well and finally, assisting in a better dispersion of CEO in the chitosan gel during the membrane formation [29].

### 3.3. UV-Vis and PL Spectrometry

#### 3.3.1. UV-Vis Spectrometry

The films were light yellow from the interaction between chitosan and ZnO or Ag NPs and CEO. The absorption spectra for the samples CZ1–CZ4 are presented in Figure 4 (left). The spectra present a broad absorption band in the UV region, which is caused mainly by the ZnO presence, but with some minor modifications generated by the presence of chitosan. The maximum is shifting from 335 nm to 361 nm as the ZnO NPs content increases. The strong absorption band from the UV region 335–361 nm is due to the presence of polydisperse ZnO nanoparticles [41]. Also, a small absorption band at 656 nm is visible due to interactions of citronella essential oil components with ZnO nanoparticles.

The absorption spectra for the samples CA1–CA4 are presented in Figure 4 (right). The absorption spectra present a broad, intense band in the UV region, with a maximum between 336–361 nm, with a tail towards the visible part. The small absorption band at 666 nm can be attributed to interactions of citronella essential oil components with AgNPs nanoparticles. The band is bathochromic shifted when compared to ZnO containing films.

The absorption spectra for the samples CZA1–CZA5 are presented in Figure 5. The absorption spectra are more complex than the spectra of the previous samples, indicating a higher interaction degree between the components. In the UV region, an intense absorption band with maxima between 334–363 nm is presented due to the ZnO content of the samples. This absorption band is in direct correlation with the ZnO content of the films. A similar increase of spectral complexity was also reported for chitosan/TiO_2_/Ag NPs (ZnO and TiO_2_ have similar spectra). The absorption of UV and visible light by ZnO is enhanced by the presence of both chitosan and AgNPs [54]. The increase of ZnO absorbance in the visible domain, in the presence of AgNPs, is also reported in the literature [41].

In the violet region of the spectrum, a second intense band can be observed at 432 nm, which can be attributed to the surface plasmon resonance (SPR) of AgNPs [54,55,56]. The SPR peak is usually in domain 400–600 nm, depending on the size and shape of AgNPs [56]. Similar spectra with asymmetrical peaks were reported for ZnO&AgNPs in literature [41], with a single SPR peak being characteristic for spherical AgNPs. The bathochromic shift of this band is due to the size of Ag NPs (~30 nm) [43]. The small absorption band at 660 nm can be attributed to interactions of citronella essential oil components with both nanoparticles, with the peak being situated between wavelengths observed in CZ and CA films.

In conclusion, the films have good absorption in the UV region, which makes them suitable as effective UV barriers. The transmittance for CZA3 is under 20% in the 300–380 nm interval, while for CZA1, where UV absorption was lower, it still has a transmittance under 30% in the 300–400 nm range. The UV barrier property of packaging films is important since blocking higher energy photons can retard lipid oxidation and preserve the organoleptic properties of the packaged food, leading to a longer shelf life [53]. All the composite films presented lower transmittance in the visible domain when compared with control chitosan film as the incorporation of nanoparticles and CEO reduced the transparency. This is similar to other composites with high nanoparticle content reported in the literature [50]. The light barrier is also important in food preservation in order to avoid photo-oxidation of organic compounds and degradation of pigments, vitamins, and other nutrients [57]. The opacity values calculated as −logT_600_/*x*, where T_600_ is the fractional transmittance at 600 nm and *x* is the film thickness in mm, gave values between 12.51–18.37. These values indicate that the films are rather opaque, with low transparency due to nanoparticle content.

#### 3.3.2. PL Spectrometry

The fluorescence spectra of CZ1–CZ4 films are presented in Figure 6 (left). The fluorescence bands of chitosan (~395 nm) and ZnO nanoparticles (~425–450 nm) are combined into a resulting broad, intense single emission band, in the range 400–450 nm, with an asymmetrical tail towards the blue region (500 nm) [58]. At low ZnO content (CZ1 and CZ2 films), the intensity of the fluorescence band of chitosan is enhanced, but a further increase of the ZnO ratio to chitosan decreases the intensity of the violet emission band. For the CZ3 and CZ4 samples, the asymmetry of the gaussian is becoming more evident.

The fluorescence spectra of CA1–CA4 films are presented in Figure 6 (right). The fluorescence band appear at 402 nm, is attributed to the chitosan. Even for CA1 sample, which has the lowest Ag NPs content, the band is over two times more intense than for simple chitosan film. The increasing Ag NPs content of CA4 leads to the enhancing of the violet emission of the sample, more than three times when compared with control chitosan sample. The band has an asymmetrical tail towards blue region (500 nm).

The fluorescence spectra of CZA1–CZA5 films are presented in Figure 7. The fluorescence band appear at 397 nm and it is asymmetrical with a tail towards blue and green regions (500–550 nm). Low quantities of ZnO nanoparticles and AgNPs enhance the chitosan fluorescence CZA1 films having the higher emission intensity [59]. As ZnO nanoparticles ratio increases, the fluorescence of chitosan is quenched, CZA3 film having the lowest intensity fluorescence. The emission is also dependent on AgNPs content, but contrary on the results obtained for CA films. Increasing the AgNPs content of film will lower the fluorescence intensity, while the lower content in CZA4 film generate a higher intensity of the emission band. This behavior indicates, like the UV-Vis spectra, the presence of more complex interactions between chitosan and nanoparticles than in CZ or CA films.

### 3.4. Thermal Analysis

Thermal analysis curves for chitosan and CZ1–CZ4 films are presented in Figure 8 and relevant data are presented in Table 3. As expected, the increased proportion of ZnO leads to a higher thermal stability of the films. In the first degradation phase, up to ~200 °C, the samples will be dehydrated and volatile compounds from essential oil will be removed [60].

The degradative-oxidative processes of organic part take place after 250 °C when exothermic effects are presented on DSC curves, low molecular weight fractions, and side chain moieties being eliminated. Over 400 °C, the carbonaceous residue is oxidized, with the process being accompanied by a strong, sharp exothermic effect. At the same time, due to its inert nature vs. oxidative atmosphere, the residual mass is entirely composed from ZnO and therefore, the highest value is for CZ4 film.

Comparing the temperatures at which the samples lost 10–50% of initial mass, this indicates a better stability for CZ4 film for all temperature intervals. CZ1 is slightly more stable than CZ2 (usually 13 °C shift for all the T20–T50 and with 33 °C for T10). The lowest stability was observed for CZ2 film, most probably due to the porous structure as presented in SEM analyses. The many small pores allow better heat transfer in the whole membrane and in this case, the protective role of the ZnO is limited. In the case of CZ3 and CZ4, even if some cracks are present, preferential heat transfer occurs through them, but in the other part of the membrane, the protective effect of the ZnO appears. If one compares the temperatures corresponding to the mass loss of 20–50%, they can conclude that CZ4 is much more stable and a mass loss of 50% appears at a higher temperature (of at least 100 °C) compared with CZ1–CZ3. The high stability of the CZ films can be explained by a higher nanoparticle content as the literature reports that chitosan-ZnO films decompose rather quickly, 96% at 280 °C when ZnO content is low (~2.5%) [50], but with good stability for chitosan: ZnO 1:2 ratio [61]. The main reason for this is the strong interactions between ZnO NPs and chitosan side groups, hindering degradation of the polymeric support as indicated in [26,49].

Thermal analysis curves for CA1–CA4 films are presented in Figure 9 and relevant data are presented in Table 4. In the first degradation phase, up to ~140 °C, the samples will be dehydrated and volatile compounds from essential oil will be removed. The degradative-oxidative processes of organic part take place after 200 °C when exothermic effects are presented on DSC curves. The low content of AgNPs does not have a direct impact on the residual mass, but helps lowering the carbonaceous residuals of chitosan, acting like a catalyst. Increasing the AgNPs content will also increase the melting point of the films as they better absorb the energy [52].

At lower temperatures, CA3 presents the best stability (higher T_10_ value) but after that, CA2 has the highest values for T_20_–T_50_. In the end, at T_50_, the values are fairly equal.

Thermal analysis curves for the CZA1–CZA5 films are presented in Figure 10 and relevant data are presented in Table 5. In the first degradation phase, up to ~220 °C, the samples will be dehydrated and volatile compounds from essential oil will be removed. The degradative-oxidative processes of organic part take place after 250 °C when exothermic effects are presented on DSC curves. The CZA4 film, with lower AgNPs content behaves somewhat differently, indicating that it has retained more water. The organic part is completely degraded in the interval 400–460 °C, with the process being accompanied by a very strong exothermic effect. As expected, increasing the ZnO content from CZA1 to CZA3 will lead to a higher residual mass. The highest thermal stability was observed for the CZA3 composition, which has T_10_ = 231 °C. In fact, except the CZA4, all other films had T_10_ values over 200 °C, which indicates a better thermal stability compared with chitosan-ZnO or chitosan-AgNPs films.

### 3.5. Scanning Electron Microscopy (SEM)

The SEM images are useful for analyzing the film’s surface morphology. The SEM images for the chitosan control film present an uneven surface, with many crystalline deposits. Higher magnifications indicate that chitosan flakes have crystalized in random directions.

For the first lot, for the films obtained from chitosan-ZnO-CEO, the SEM images are presented in Figure 11. The SEM images indicate the porous structure obtained for CZ1 film, with a homogenous distribution of micropores ranging from 5 to 50 μm. This most likely happened, because the added ZnO can be glued by the chitosan polymeric chains and generate a porous structure, even if the pure chitosan film has a pore-free structure. The pore size decreases sharply for the CZ2 sample to a diameter smaller than 5 μm, but also with a highly homogeneous distribution. Some random pores can also be observed for CZ3 or CZ4 films, but in these cases, cracks are also visible, especially for CZ4. This porous structure can be expected to present higher water vapor and gases permeability, but in fact, the permeability decreases until CZ2 and after this, it starts to increase. This can be explained by the fact that the pores are not connected and it does not cross the membrane, therefore in this case, the density of the membrane slightly increases. The higher values are obtained for CZ3 and CZ4, where the cracks permit a better penetration of the film structure. The higher magnification SEM images (20 k) reveals a homogeneous distribution of ZnO nanoparticles in the chitosan film structure.

For the second lot, obtained from chitosan-AgNPs-CEO, the SEM micrographies are presented in Figure 12. The SEM images indicate the flakes structure of the chitosan films. For CA1, the highly irregular flakes seldom appear connected, forming a rugged surface in which AgNPs agglomeration can be identified. For CA2 film, the flakes appear larger, but well connected between them, resembling a fish scales arrangement. For CA3, the pore dimensions are smaller and the film looks more homogenous than CA1 or CA2. Finally, for CA4, the film presents a smooth surface, with no visible cracks or pores, with some AgNPs agglomerations. According to these data, the permeability through these membranes is expected to decrease gradually [62].

The SEM images for the films containing both types of nanoparticles (ZnO and Ag) are presented in Figure 13. The surface of the films is homogenous, but with clear differences as ZnO proportions increases. While for CZA1, ZnO seems well dispersed in the film mass and highlights, similar to CZ2, pores (these two membranes having the same content of ZnO), for CZA2, some ZnO nanoparticles deposits on the surface can be noticed. For CZA3, the ZnO nanoparticles agglomeration increases and the surface of the film becomes rugged, with lots of pores that seem to be deep in the membrane. Comparing the CZA4 film with CZA2, this indicates that lower AgNPs content promoted some platelet formation and their aggregation. The CZA5 film has a homogenous surface with microbubbles like structures. Some nano-size cracks are visible in the structure of the films as can be seen at a magnification of 100,000. According to these micrographies, it is expected that water vapor permeation decreases for all the samples with smooth surfaces and only nanosized cracks, which most probably are not crossing the membrane.

### 3.6. Water Vapor Permeability (WVP)

Packaging films for food should ideally insulate it from loss of flavor, water, and other specific substances. WVP is important as a control for moisture transfer between food and the outside environment. Usually a high WVP can lead to microbial spoilage of food. The results of WVP testing are presented in Table 6. Important decreases in WPV are observed for almost all samples when compared with chitosan control film. The smaller decreases observed for CZ3, CZ4, and CZA3 are more likely because of some cracks in the polymer matrix, promoted by increased ZnO content.

As the content of ZnO nanoparticles increased, the WVP of samples decreased. As the nanoparticles are water resistant when compared with the polymer matrix, the decrease of WVP can be ascribed to the formation of new denser and more tight structures between chitosan and nanoparticles. Therefore, the water must pass a more complex/longer pathway as presented in Figure 14. A porous structure can be expected to present higher WVP, but only if the pores are connected. As the SEM images indicate, the CZ1 and CZ2 films have a porous surface, but this is not connected in the body of the film, so the pores are not connected and do not cross the membrane; therefore, in this case, the WVP values are better than for the control film.

The AgNPs are capped with PVP and their presence has a similar effect as ZnO nanoparticles, decreasing the WVP. As their tendency of agglomeration is lower, increasing the AgNPs content in chitosan films does not lead to cracks promotion and therefore, the WVP decrease linearly in this case, with the CA series having the more predictable behavior and the less significant change compared with the chitosan control film. The presence of CEO also has the same result due to its hydrophobic nature.

### 3.7. Antimicrobial Activity

The tests were performed against three strains: one Gram-positive, one Gram-negative, and a fungal strain. While both bacterial strains, *S. aureus* and *E. coli*, can cause severe foodborne disease, *C. albicans* is the most frequent cause of mycoses and is ranked in the top five nosocomial pathogens [63]. The antimicrobial activity of the samples is dependent on the nature of the antimicrobial agent, the composition, as well as the nature of the fungal/bacterial strain. While the antibacterial activity of ZnO is well known [64,65,66], the antifungal activity of ZnO nanoparticles has not received the same attention until recently as there are still a limited number of studies that have been carried out [67,68]. The mechanism of antibacterial activity for ZnO is complex and includes ROS generation, particle internalization, and cell membrane damage [69].

The ZnO NPs were found to be more efficient against *E. coli* than *S. aureus*. Similar reports are presented in the literature [31,70]. However, results depend strongly on the size and shape of nanoparticles. While in [31] ZnO NPs with a concentration under 100 mg/mL exhibited no activity, in [70], antibacterial activity was obtained for concentrations of 10 mg/mL. As expected, increasing the ZnO NPs content of CZ films will enhance the antimicrobial activity. Our results are similar with those reported in [26,57]. However, different results are reported in [71], probably because the ZnO particles used have dimensions over 100 nm and tend to form agglomerates.

According to the antimicrobial assessments, one can conclude that all series were more active against *E. coli* followed by *S. aureus* and *C. albicans*. In the case of the CZ series (Figure 15), the highest antimicrobial activity was observed for CZ3 and CZ4, which can be correlated with a higher ZnO NPs content. The increasing quantity of ZnO from CZ3 to CZ4 does not lead to a significant improvement in antibacterial activity, but enhanced the antifungal activity.

Although the ZnO does not have a remarkable antifungal activity and neither does chitosan, the films from CZ1 to CZ4 indicate an increase inhibition over *C. albicans*. CEO is well known for its antifungal activity (Figure 15), but all the samples have the same amount of essential oil. Therefore, the increased antifungal activity from CZ1 to CZ4 cannot be directly attributed to CEO, but rather to the increased concentration of ZnO NPs. As ZnO alone exhibited a rather low antifungal activity, our hypothesis is that ZnO NPs are sensitizing the fungal cells and thereafter, CEO can be more effective against *C. albicans*. This suggest the existence of synergic activity for the three components, chitosan-ZnO-CEO.

In the CA series, the antimicrobial activity is lower compared with the CZ series, probably due to a lower load of nanoparticles (Figure 16).

From the bacterial strains, *S. aureus* is more susceptible to the presence of AgNPs [43]. Nevertheless, the activity of CA films was higher against the *E. coli* strain. The higher resistance of *S. aureus* is probably due to differences in the intracellular antioxidant content [31]. The best antibacterial activity for the CA series was obtained for CA3 film, further increasing AgNPs load and decreasing the antibacterial activity, probably due to some agglomeration of NPs. At the same time, antifungal activity increased only marginally from CA3 to CA4. The antifungal activity was higher as the AgNPs content increased, with CA4 being the most active in the series against *C. albicans*.

The CZA series is the most potent against the tested strains, probably due to the synergic activities of all the components (Figure 17). The excellent antimicrobial results of CZA films are due to the multiplication of microorganisms inactivation pathways, such as electrostatic interaction, reactive oxygen species generation, membrane rupture and internalization of nanoparticles, SPR effect of AgNPs, and CEO antimicrobial activity. Synergic antimicrobial activities were also reported in the literature by [40,41,54]. The strongest antimicrobial activity was exhibited by the CZA3 membrane, with the inhibition diameter being ~30 mm for *S. aureus* and *E. coli* and over 20 mm for *C. albicans*, which means strong antimicrobial activity. By comparing CZA3 with the CZ4 (both films have the same content of ZnO and CEO, but in the former, there are also AgNPs), the increase of antibacterial activity is high for *S. aureus* and moderate for *E. coli*. The addition of AgNPs did not further improve the antifungal activity of CZA3 vs. CZ4 as this was already high. Comparing the CZA1–CZA3 series in which only the ZnO content increases, we can see a direct correlation between ZnO content and antimicrobial activity. Against *C. albicans*, the inhibition diameter increases three times, while for *S. aureus*, the diameter doubles. By comparing the inhibition diameters with the corresponding samples CZ2–CZ4, the influence of the AgNPs can be identified against both bacterial strains. At the same time, comparing the samples CZA4, CZA2, and CZA5, where only the AgNPs content varies, we can observe the higher impact against the fungal strain as it was also noticed for the films CA1–CA4.

All obtained samples, except CZA3, presented a better activity against *E. coli* compared to *S. aureus*. Similar results were reported in [41] when the polymer used was carboxymethyl cellulose. When the support was 4-styrenesulfonic acid and maleic acid, the results were better for *S. aureus*. In other studies, the ZnO & AgNPs simple mix [39] or deposited on activated carbon [40] were found to be more active against *S. aureus* compared to *E. coli*. This suggests that the polymeric support can also play an important role.

The high antimicrobial activity observed for CZA films makes them a good candidate for packaging materials for fruits. Similar chitosan-AgNPs films [72,73] are reported in the literature as coatings for peaches and red grapes, with an extended shelf life of up to 14 days. Also, for chitosan–ZnO films, the literature reports extended shelf life for green grapes [28].

### 3.8. Preliminary Evaluation of CZA1–CZA5 Films as Coatings for Grapes

The chitosan films with ZnO&AgNPs loaded with citronella essential oil were used for packing white grapes that were stored at room temperature (30 °C and 60% relative humidity) for 14 days. A control lot was packaged in polyethylene film. The preliminary visual quality check (Figure 18) suggests that CZA1–CZA5 films were effective in preserving the grapes. The decaying of the control sample was total, with a clear mildew appearance, moldy areas on the surface, and sticky liquid leaking from the fruits.

The grapes stored in CZA films presented only a few rusty spots on the surface of the samples, which did not evolve or damage the fruits. The visual appearance of the samples packaged in CZA3 film is the most acceptable after 14 days. The results suggest that CZA films could be a promising coating for white grapes, which requires further study.

## 4. Conclusions

A new antimicrobial film based on chitosan was obtained. The film was modified by adding ZnO and AgNPs loaded with citronella essential oil to enhance the antimicrobial properties. To better highlight the synergism of the antimicrobial agents, the simple chitosan-ZnO-CEO and chitosan-Ag-CEO films were obtained and also tested. The three series were characterized and finally evaluated as antimicrobial membranes for food packaging. According to the results of the antimicrobial assay, we can clearly conclude that the combination of chitosan, CEO, ZnO, and AgNPs is presenting a potent antimicrobial activity, also bringing some enhancements in terms of water vapor barrier properties and UV barrier. The results in the CZ series suggest that ZnO NPs (with low antifungal activity) are sensitizing the fungal cells, thereafter helping CEO to be more potent against *C. albicans*. Results from the CZA series indicate that increasing the load of AgNPs further improves the antifungal activity.

The obtained results suggest that the synergism of all components can assure strong antimicrobial activity against the fungal (*C. albicans*) and bacterial (*S. aureus* and *E. coli*) strains, with the inhibition diameter reaching ~30 mm against the two bacterial strains and over 20 mm for the fungal strain. It can be concluded that this kind of membrane has a broad-spectrum antimicrobial activity and can be potentially used as coatings for fruits, with the CEO scent also being compatible with such foodstuff.

## Figures and Tables

**Figure 1 foods-09-01801-f001:**
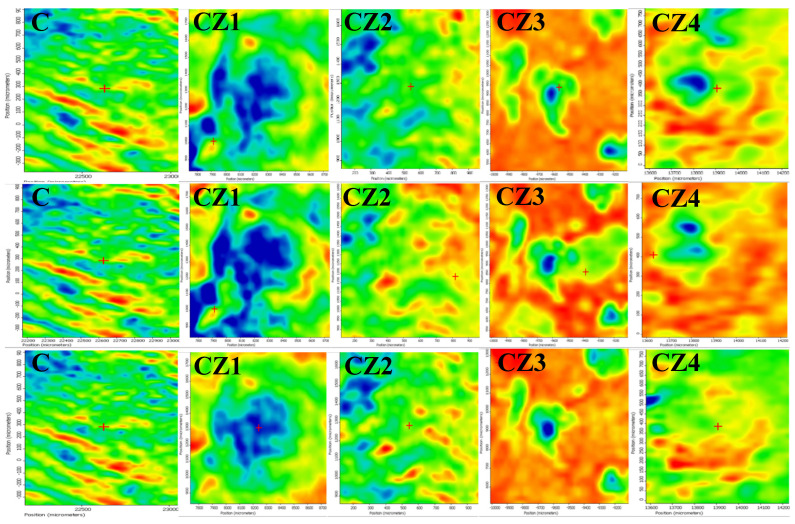
The FTIR maps for CZ1–CZ4 films at wavenumbers 3275 cm^−1^ (**top row**); 1743 cm^−1^ (**intermediate row**); and 1643 cm^−1^ (**bottom row**).

**Figure 2 foods-09-01801-f002:**
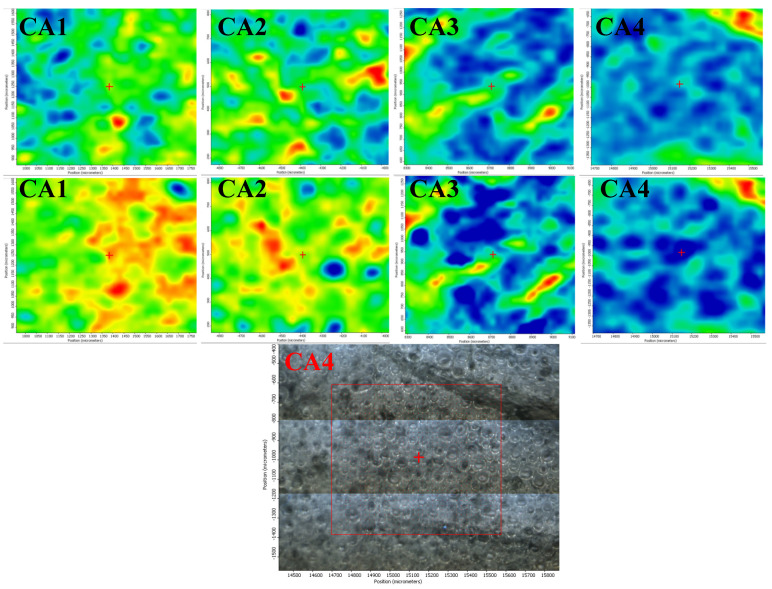
The FTIR maps for CA1–CA4 films at wavenumbers 3500 cm^−1^ (**top row**) and 1740 cm^−1^ (**intermediate row**); the last row represents the FTIR video image highlighting the morphology of the sample CA4 (in fact, all these membranes have the same aspect).

**Figure 3 foods-09-01801-f003:**
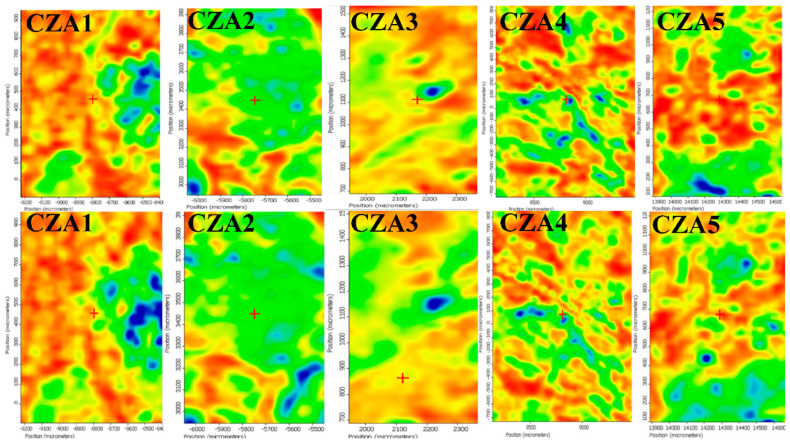
The FTIR maps for CZA1–CZA5 films at wavenumbers 3393 cm^−1^ (**top row**) and 1643 cm^−1^ (**bottom row**).

**Figure 4 foods-09-01801-f004:**
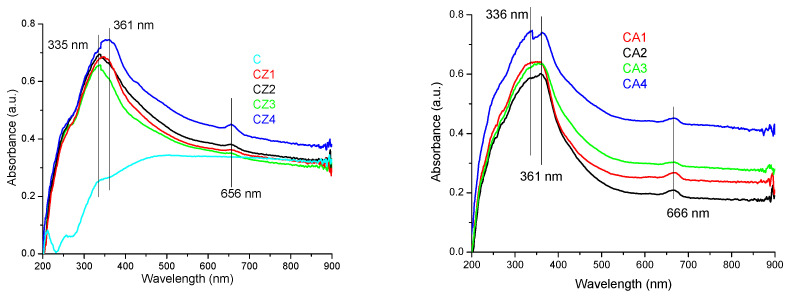
The UV-Vis spectra for chitosan and CZ1–CZ4 films (**left**) and CA1–CA4 films (**right**).

**Figure 5 foods-09-01801-f005:**
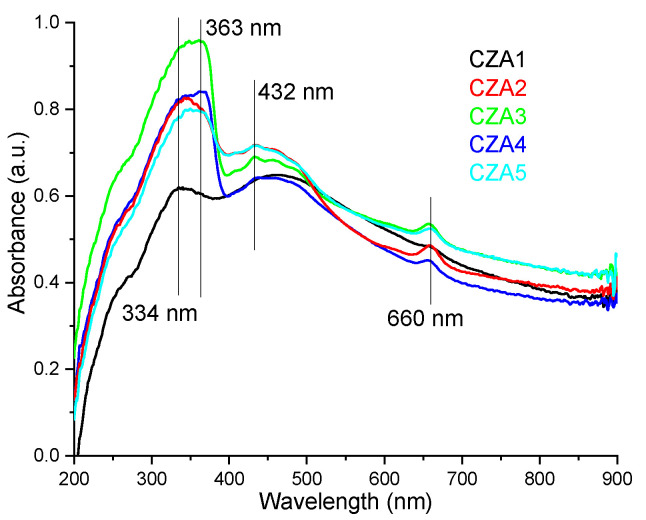
The UV-Vis spectra for CZA1–CZA5 films.

**Figure 6 foods-09-01801-f006:**
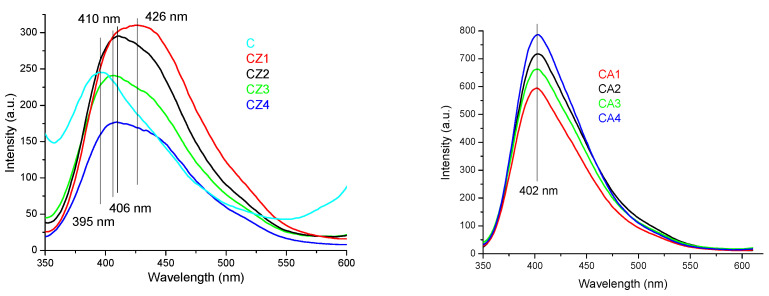
The PL spectra for chitosan and CZ1–CZ4 films (**left**) and CA1–CA4 films (**right**).

**Figure 7 foods-09-01801-f007:**
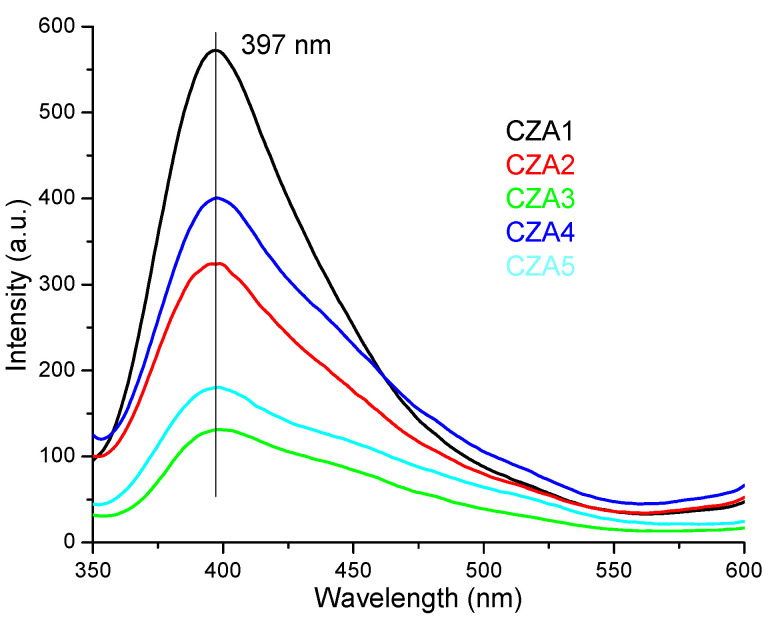
The photoluminescence spectrum (PL) spectra for chitosan and CZA1–CZA5 films.

**Figure 8 foods-09-01801-f008:**
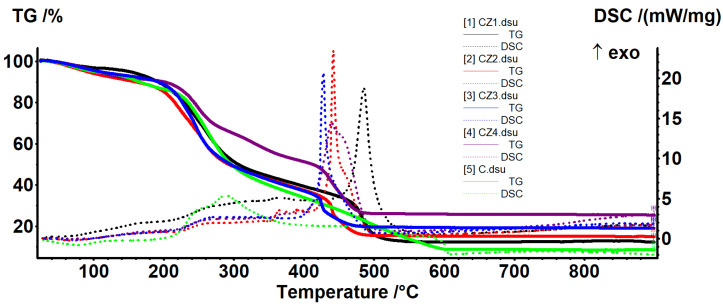
The TG-DSC curves for chitosan and CZ1–CZ4 films.

**Figure 9 foods-09-01801-f009:**
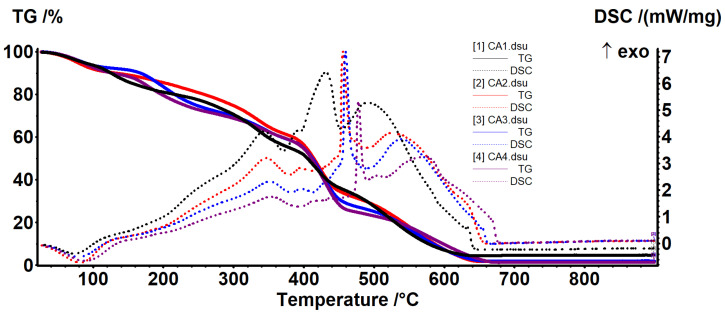
The TG-DSC curves for CA1–CA4.

**Figure 10 foods-09-01801-f010:**
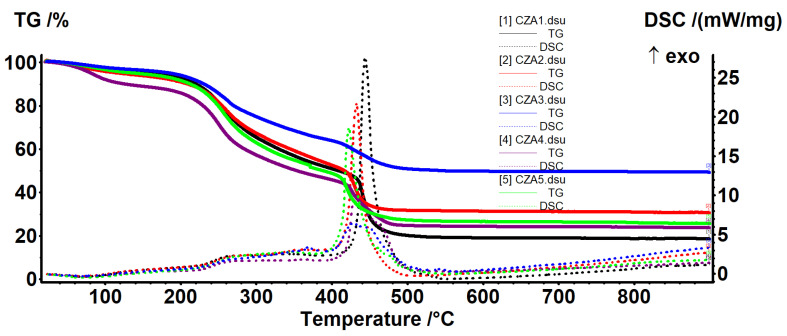
The TG-DSC curves for CZA1–CZA5.

**Figure 11 foods-09-01801-f011:**
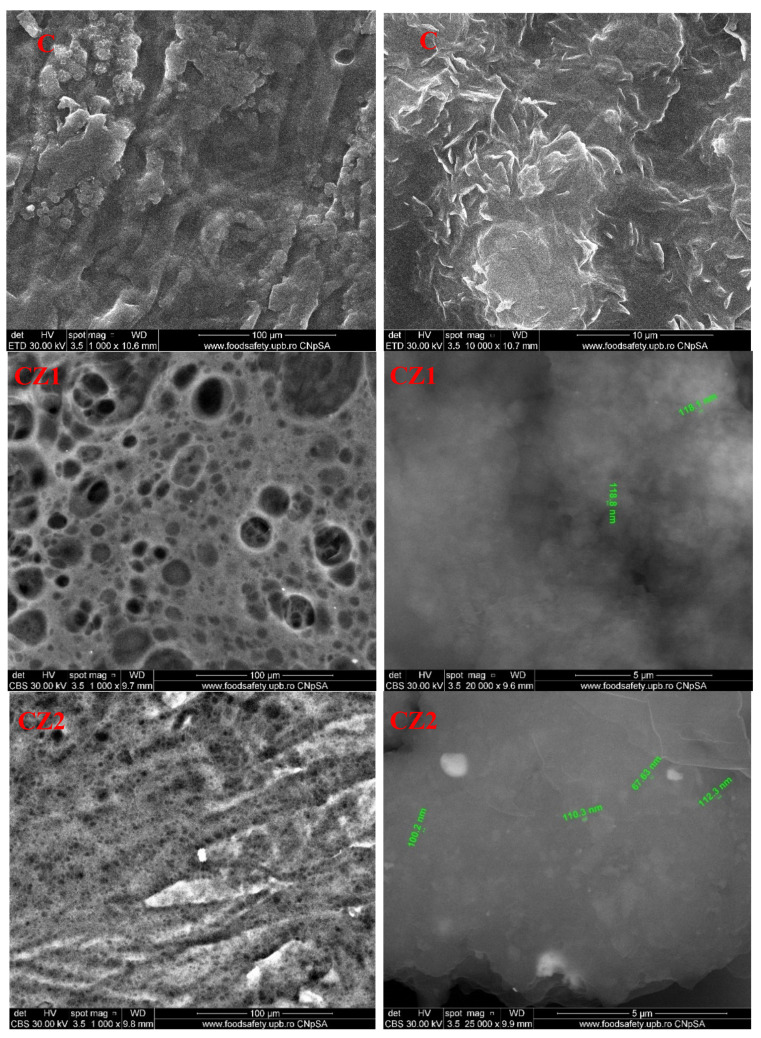
The SEM images for chitosan, CZ1, CZ2, CZ3, and CZ4 films.

**Figure 12 foods-09-01801-f012:**
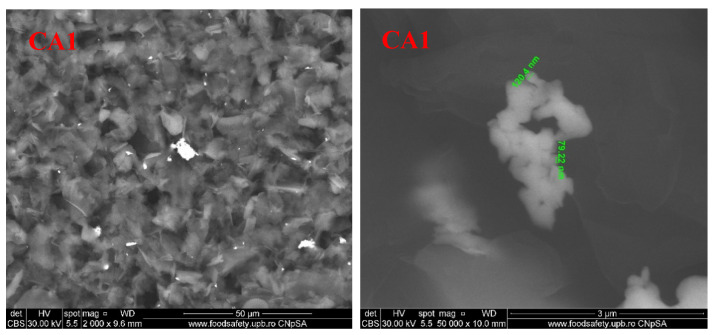
The SEM images for chitosan, CA1, CA2, CA3, and CA4 films.

**Figure 13 foods-09-01801-f013:**
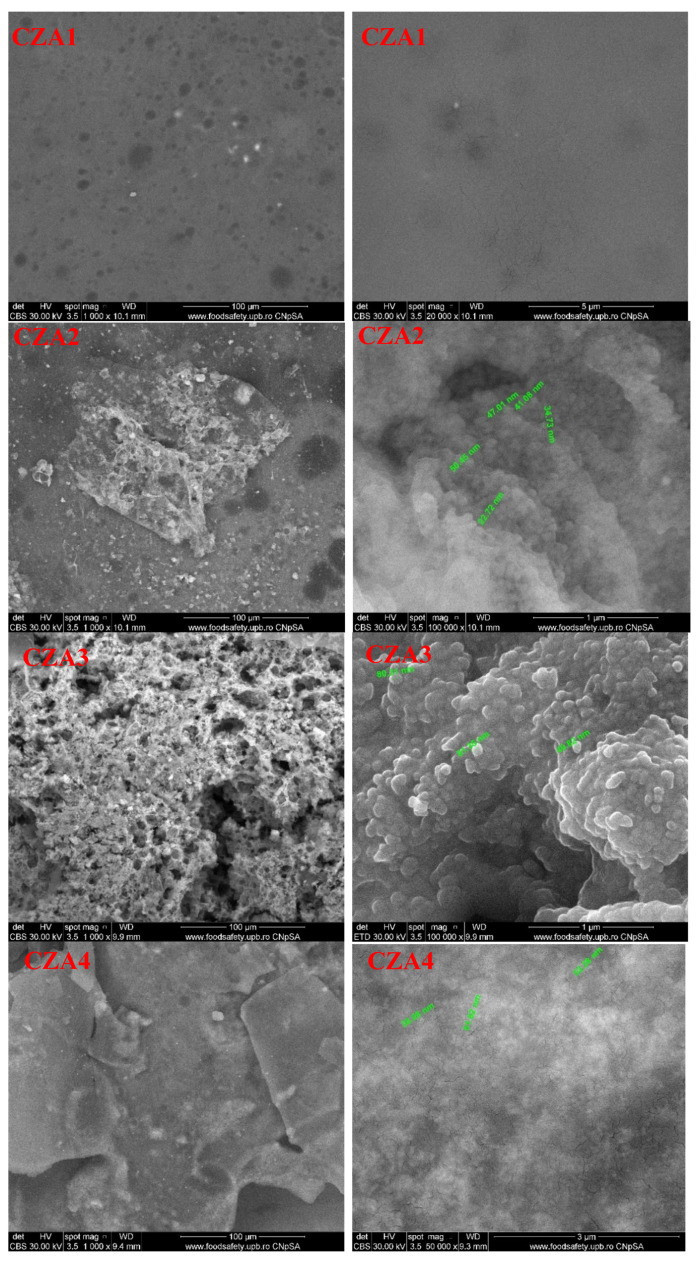
The SEM images for chitosan, CZA1, CZA2, CZA3, CZA4, and CZA5 films.

**Figure 14 foods-09-01801-f014:**
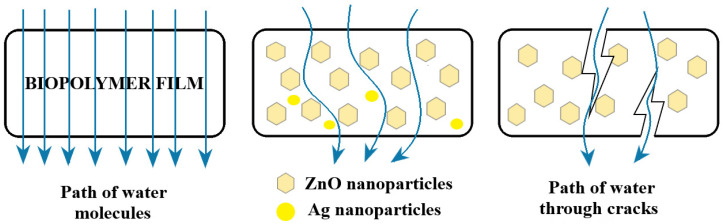
The water vapor permeability (WVP) mechanism for the control film (**left**), chitosan–ZnO&AgNPs films (**middle**), and cracked films (**right**).

**Figure 15 foods-09-01801-f015:**
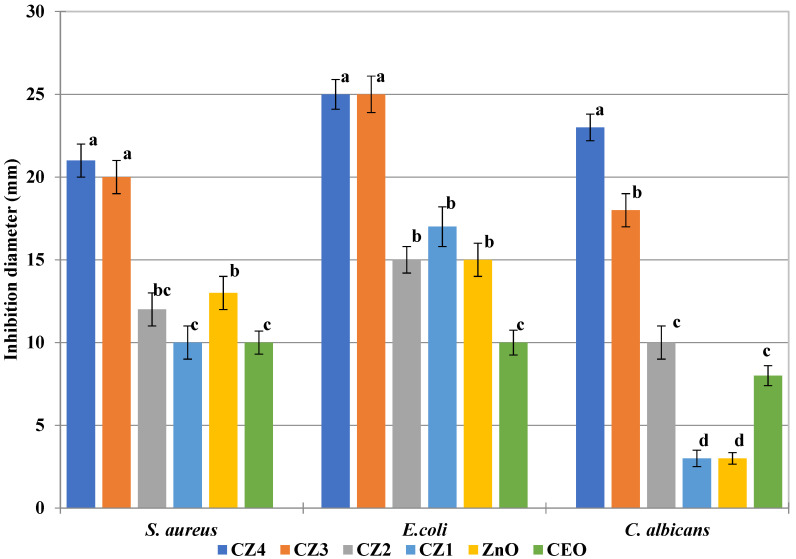
The antimicrobial activity of CZ1–CZ4 films and ZnO nanoparticles—different letters indicate statistically significant differences between films (*p* < 0.05).

**Figure 16 foods-09-01801-f016:**
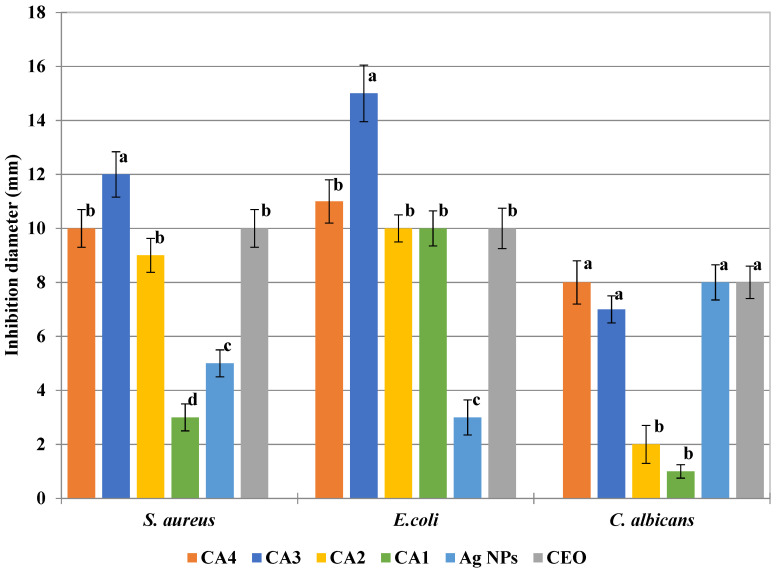
The antimicrobial activity of CA1–CA4 films and CEO—different letters indicate statistically significant differences between films (*p* < 0.05).

**Figure 17 foods-09-01801-f017:**
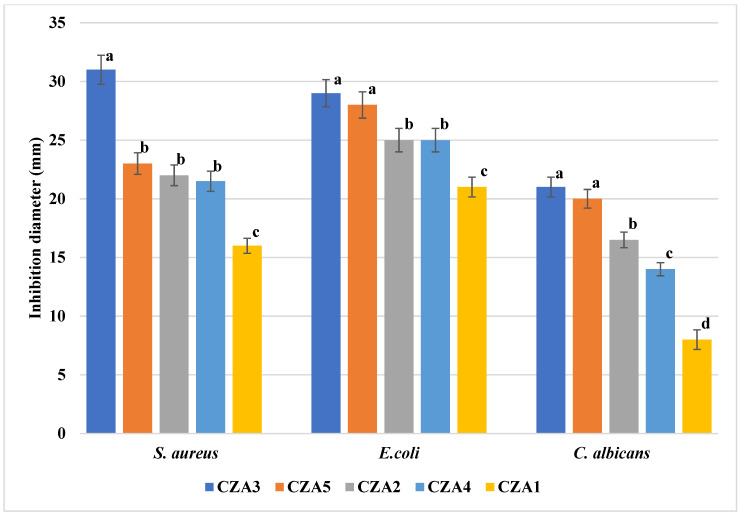
The antimicrobial activity of CZA1–CZA5 films—different letters indicate statistically significant differences between films (*p* < 0.05).

**Figure 18 foods-09-01801-f018:**
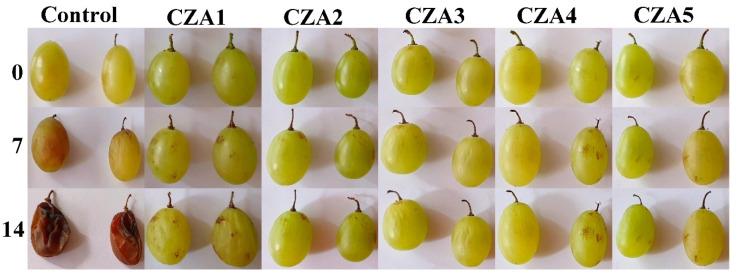
Visual appearance of white grapes packaged in control (polyethylene film) and CZA1–CZA5 films, after 0, 7, and 14 days storage at 30 °C and 60% relative humidity.

**Table 1 foods-09-01801-t001:** The chitosan–nanoparticles–citronella essential oil (CEO) films composition.

Sample Code	Chitosan (g in 100 mL of 1% (*v/v*) Acetic Acid)	ZnO NPs (g in 15 mL Water)	Ag NPs (mL Solution)	CEO (mL)
CZ1	0.66	0.165	-	1
CZ2	0.66	0.33	-	1
CZ3	0.66	0.66	-	1
CZ4	0.66	0.99	-	1
CA1	0.66	-	10	1
CA2	0.66	-	15	1
CA3	0.66	-	20	1
CA4	0.66	-	30	1
CZA1	0.66	0.33	15	1
CZA2	0.66	0.66	15	1
CZA3	0.66	0.99	15	1
CZA4	0.66	0.66	10	1
CZA5	0.66	0.66	30	1

**Table 2 foods-09-01801-t002:** Assignment of relevant IR absorption bands of chitosan control and chitosan/NPs/CEO films.

Sample	υZn-O	As. C-O-C	Amide III Band (υC-N; δN-H)	Amide II Band (υC-N; δN-H)	Amide I Band (υC-N; υC=O)	C=O Group of CEO [22]	υC-H (sat)	υO-H υN-H
C	-	1152	1318	1553	1651	-	2921	3277
CZ1	426 473	1152	1338	1558	1643	1745	2918	3255
CZ2	427 474	1152	1338	1559	1643	1747	2920	3263
CZ3	428 467	1152	1338	1557	1643	1744	2919	3295
CZ4	428 467	1152	1338	1558	1643	1747	2922	3305
CA1	-	1152	1317	1558	1647	1746	2921	3368
CA2	-	1154	1317	1563	1647	1746	2922	3392
CA3	-	1153	1317	1560	1645	1745	2920	3393
CA4	-	1155	1317	1560	1644	1744	2920	3394
CZA1	419 475	1152	1339	1559	1643	1744	2918	3254
CZA2	421 472	1153	1329	1559	1643	1745	2921	3302
CZA3	426 469	1152	1338	1555	1643	1744	2924	3297
CZA4	419 468	1153	1339	1559	1643	1746	2922	3304
CZA5	429 470	1153	1340	1559	1647	1747	2918	3292

**Table 3 foods-09-01801-t003:** Temperatures at which chitosan and CZ1–CZ4 films lost 10%, 20%, 30%, 40%, or 50% of initial mass.

Sample	T_10_	T_20_	T_30_	T_40_	T_50_	Residual Mass
CZ1	192 °C	229 °C	252 °C	273 °C	306 °C	12.23%
CZ2	159 °C	216 °C	239 °C	260 °C	293 °C	14.86%
CZ3	192 °C	227 °C	244 °C	260 °C	295 °C	18.91%
CZ4	200 °C	244 °C	270 °C	330 °C	412 °C	25.33%

**Table 4 foods-09-01801-t004:** Temperatures at which CA1–CA4 films lost 10%, 20%, 30%, 40%, or 50% of initial mass.

Sample	T_10_	T_20_	T_30_	T_40_	T_50_	Residual Mass
CA1	125 °C	215 °C	303 °C	348 °C	405 °C	4.65%
CA2	130 °C	258 °C	326 °C	384 °C	414 °C	1.89%
CA3	167 °C	216 °C	297 °C	368 °C	412 °C	1.87%
CA4	126 °C	197 °C	286 °C	369 °C	411 °C	1.24%

**Table 5 foods-09-01801-t005:** Temperatures at which CZA1–CZA5 films lost 10%, 20%, 30%, 40%, or 50% of initial mass.

Sample	T_10_	T_20_	T_30_	T_40_	T_50_	Residual Mass
CZA1	220 °C	254 °C	281 °C	330 °C	408 °C	18.47%
CZA2	210 °C	255 °C	287 °C	346 °C	417 °C	30.64%
CZA3	231 °C	270 °C	337 °C	427 °C	542 °C	49.27%
CZA4	126 °C	230 °C	254 °C	285 °C	354 °C	23.72%
CZA5	213 °C	250 °C	273 °C	316 °C	389 °C	25.67%

**Table 6 foods-09-01801-t006:** Water vapor permeability (WVP) for control, CZ, CA, and CZA films.

Film Code	WVP (10^−10^ g/Pa∙m∙s)
C (chitosan control)	1.411 ± 0.011 ^a^
CZ1	1.214 ± 0.018 ^a,b,c,d^
CZ2	1.165 ± 0.023 ^b,c,d^
CZ3	1.298 ± 0.056 ^a,b^
CZ4	1.315 ± 0.124 ^a,b^
CA1	1.326 ± 0.078 ^a,b^
CA2	1.301 ± 0.081 ^a,b^
CA3	1.278 ± 0.056 ^a,b,c^
CA4	1.262 ± 0.035 ^a,b,c^
CZA1	1.143 ± 0.091 ^b,c,d^
CZA2	1.055 ± 0.078 ^c,d^
CZA3	1.281 ± 0.137 ^a,b,c^
CZA4	1.117 ± 0.069 ^b,c,d^
CZA5	1.026 ± 0.086 ^d^

Different superscript letters indicate statistically significant differences between films (*p* < 0.05).

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
