# Peer review of "Innovative Antimicrobial Chitosan/ZnO/Ag NPs/Citronella Essential Oil Nanocomposite—Potential Coating for Grapes"

_foods, 2020, doi:10.3390/foods9121801_

Round 1
Reviewer 1 Report
The work of Motelica et al., propose an innovative chitosan-based coating with broad antimicrobial activity for application in the food industry. A comprehensive physico-chemical characterization of the coatings has been performed, as well as an in vitro analysis of the antimicrobial activity against three strains representative of Gram- negative and Gram-positive bacteria and fungi.
In my opinion this study only partially fit within the aims of the journal, since no application of the proposed coatings has been investigated in a food model. A corresponding experimental section should be required.
Title. Change “antibacterial” to “antimicrobial”. No evidence has been reported on the application of the coatings for fruits
Section 2.5. The conditions of the inoculum should be specified. From cryoconserved stock? From overnight cultures? At which concentration? Which media was used for growth?
The authors, also, should explain why they chose Candida as model for fungi, since several fungal genera (i.e. Aspergillus, Penicillium, Botrytis, Fusarium, …) are responsible for the main spoiler events in the food industry.
Statistical analysis must be added. Otherwise is a no-sense discuss significant-differences in the results. Significant differences must be indicated also in Figures 16-18.
Section 3.7. Why has not been investigated the antimicrobial activity of Ag, as done for ZnO and CEO?
Figures 16,17,18. y-ax. Specify the unit of the inhibition diameter
L418. Change “indicate” to “suggest”
L430. Change “because” to “probably due”
Results in general, and antimicrobial activity in particular, are poorly discussed.
Aureus, Albicans, Coli are always lowercase. Please, check throughout all the manuscript.
Author Response
Point 1: Title. Change “antibacterial” to “antimicrobial”. No evidence has been reported on the application of the coatings for fruits
Response 1: We are thankful for pointing out this weakness. The title was changed in “Innovative antimicrobial chitosan/ ZnO/ Ag NPs /citronella essential oil nanocomposite - potential coating for fruits"
The objective of this research was to obtain a biodegradable antimicrobial film that can be used as packaging material for extending shelf life of fresh fruits. Therefore, chitosan was chosen as base biopolymer, with ZnO and Ag nanoparticles as filler / antimicrobial agents for the fabrication of nanocomposite films. The nanoparticles were loaded with citronella essential oil in order to enhance the antimicrobial activity
We started the assessment of antimicrobial activity through a screening (qualitative evaluation) on several microbial strains, and we are continuing antimicrobial tests by qualitative and quantitative methods on a much larger number of strains specific to food, on both materials, as such and put in contact with fruits (grapes and strawberries). Unfortunately, the whole study is too big to be integrated in just one article.
Point 2: Section 2.5. The conditions of the inoculum should be specified. From cryoconserved stock? From overnight cultures? At which concentration? Which media was used for growth?
Response 2: We are thankful for the suggestion to improve the manuscript. We have added detailed explanations regarding antimicrobial assessment. Inoculum was provided from microbial cell suspension that has been made in sterile physiological buffer from fresh culture with a standard density of of 1.5 X 108 CFU/mL (corresponding 0.5 McFarland standard) for S. aureus and E. coli and 3 X 108 CFU/mL (corresponding 1 McFarland standard) for C. albicans, developed on Nutrient Broth with 2% agar for bacterial strains (NBA) and Sabouraud dextrose broth with 2% agar for yeast (SA).
Point 3: The authors, also, should explain why they chose Candida as model for fungi, since several fungal genera (i.e. Aspergillus, Penicillium, Botrytis, Fusarium, …) are responsible for the main spoiler events in the food industry.
Response 3: We also included C. albicans because it is the most important cause of healthcare-associated fungal diseases (mycoses). It is also the most frequent cause of superficial and deep mycoses. In addition, antifungal resistance is reported in this species although not having the frequency and relevance of resistance in Staphylococcus, Escherichia or Pseudomonas. C. albicans is the most relevant fungal pathogen since Candida species are ranked as the fourth or fifth most common nosocomial bloodstream pathogen in the USA and some European countries, with mortality rates as high as 45% (Nanomaterials 2020, 10, 376 and Int. Microbiol. 2018, 21, 107–119).
A short phrase was added at the beginning of section 3.7.
Point 4: Statistical analysis must be added. Otherwise is a no-sense discuss significant-differences in the results. Significant differences must be indicated also in Figures 16-18.
Response 4: We are thankful for this suggestion. Statistical data were added in section 2.5. and standard deviation is indicated in figures from section 3.7. All experiments were designed and performed in triplicates, and differences were among experiments were minimal.
Point 5: Section 3.7. Why has not been investigated the antimicrobial activity of Ag, as done for ZnO and CEO?
Response 5: We are thankful for this observation. We have corrected this omission, the data for antimicrobial activity of AgNPs solution being added in fig 17.
Point 6: Figures 16,17,18. y-ax. Specify the unit of the inhibition diameter
L418. Change “indicate” to “suggest”
L430. Change “because” to “probably due”
Aureus, Albicans, Coli are always lowercase. Please, check throughout all the manuscript.
Response 6: We are thankful for pointing out these mistakes. Inhibition diameter was measured in mm. All the required corrections were made.
Point 7: Results in general, and antimicrobial activity in particular, are poorly discussed.
Response 7: We are thankful for the opportunity to enhance our manuscript. Detailed discussions and comparison with literature data were added in relevant sections (3.2.1; 3.3.1; 3.4; 3.7). Detailed discussions and comparison with literature data were also added in section 3.7., including extending shelf life results on grapes and peaches obtained with chitosan – ZnO or chitosan – AgNPs films.
Reviewer 2 Report
Although the manuscript has interesting subject and describes valuable research, in my opinion, needs to be revised.
Abstract – every time abstract should contains the most important information like most important findings and results. Some values are needed. The abstract should be reorganized.
Results and Discussion is very important part of each manuscript published. In presented manuscript, there is no discussion section. Authors should discuss their results with other scientific papers.
The conclusions should be integrated with more detailed results summarizing all the study and must reflect the innovation of this study and the perspectives.
English and style require a careful reorganization. There are a lot of language mistakes, grammatically and stylistically.
Author Response
Point 1: Abstract – every time abstract should contains the most important information like most important findings and results. Some values are needed. The abstract should be reorganized.
Response 1: We are thankful for pointing out this weakness. The abstract was enriched with relevant data from manuscript.
New packaging materials based on biopolymers are gaining an increasing attention due to many advantages like biodegradability or existence of renewable sources. Grouping more antimicrobials agents in the same packaging can create a synergic effect, resulting either a better antimicrobial activity against a wider spectrum of spoilage agents or a lower required quantity of antimicrobials. In present work we have obtained a biodegradable antimicrobial film that can be used as packaging material for food. Films based on chitosan as biodegradable polymer, with ZnO and Ag nanoparticles as filler / antimicrobial agents were fabricated by casting method. The nanoparticles were loaded with citronella essential oil in order to enhance the antimicrobial activity of the nanocomposite films. The tests made on Gram-positive, Gram-negative and fungal strains, indicated a broad-spectrum antimicrobial activity, with inhibition diameters of over 30 mm for bacterial strains and over 20 mm for fungal strain. The synergic effect was evidenced by comparing the antimicrobial results with chitosan/ZnO/CEO or chitosan/Ag/CEO simple films. According to the literature these formulations are suitable for coating for fruits. The obtained nanocomposite films presented lower water vapor permeability values when compared with the chitosan control film. The samples were characterized by SEM, fluorescence and UV-Vis spectroscopy, FTIR spectroscopy and microscopy and thermal analysis.
Point 2: Results and Discussion is very important part of each manuscript published. In presented manuscript, there is no discussion section. Authors should discuss their results with other scientific papers.
Response 2: We thank to the esteem reviewer for the opportunity to enrich the paper by comparing our research with relevant results from literature. Detailed discussions and comparison with literature data were added in relevant sections (3.2.1; 3.3.1; 3.4; 3.7). Comparison was made with relevant papers found in recent literature (23 references were added). Detailed discussions and comparison with literature data were also added in section 3.7., including extending shelf life results on grapes and peaches obtained with chitosan – ZnO or chitosan – AgNPs films.
Point 3: The conclusions should be integrated with more detailed results summarizing all the study and must reflect the innovation of this study and the perspectives.
Response 3: We are thankful for this observation. We have introduced more details and perspectives in the conclusions. We started the assessment of antimicrobial activity through a screening (qualitative evaluation) on several microbial strains, and we are continuing antimicrobial tests by qualitative and quantitative methods on a much larger number of strains specific to food, on both materials, as such and put in contact with fruits (grapes and strawberries). Unfortunately, the whole study is too big to be integrated in just one article.
The conclusion section was reworked, and relevant information were added.
Point 4: English and style require a careful reorganization. There are a lot of language mistakes, grammatically and stylistically.
Response 4: We have thoroughly checked the English language and style. Mistakes were found and corrected, some paragraphs were rephrased to become clearer and excess words were eliminated.
Reviewer 3 Report
Article review: ‘Innovative antibacterial chitosan/ ZnO/ Ag NPs /citronella essential oil coatings for fruits’
The title of the article should be changed, it is inappropriate for the presented research. ‘For fruits’? Authors did not investigate antibacterial chitosan/ ZnO/ Ag NPs /citronella essential oil coatings for fruit. Moreover, it is not known what was the purpose of the work? The purpose of the research should be specified in the Introduction section.
Other comments:
Line 68-73: Please expand on this paragraph. It's too laconic. Please present the conclusions of the research of other scientists.
Line 89-90: Provide reference to table 1 for better clarity (readability) of the text.
Table 1: What did Authors suggest when selecting levels of ZnO and Ag NPs in the CZA1-CZA5 sample?
Line 234-236: Please provide a reference to the literature for this fact.
Line 333: ‘chitosan – ZnO NPs – CEO’ it should be ‘chitosan – ZnO – CEO’
Line 405-407: In some countries there have been or are restrictions on the use of PVP in food, hence a limited number of studies may arise.
Figure 16- 18: Please remove the title from the figure and provide the unit of inhibition diameter.
Line 416-417: And what about the comparison results inhibition diameter of CZ1-CZ4 samples to the value of CEO? It was also present in the CZ1-CZ4 samples, not only ZnO.
Section 3.7 (Antimicrobial activity) and section 5 (Conclusions): Please use small letter for names: coli, aureus, albicans and enter a ‘space’ between the value and the unit of inhibition diameter.
Title of section 3 should be “Results and discussion’
Correct the number of section ‘Conclusions’ should be 4 not 5.
Author Response
Point 1: The title of the article should be changed, it is inappropriate for the presented research. ‘For fruits’? Authors did not investigate antibacterial chitosan/ ZnO/ Ag NPs /citronella essential oil coatings for fruit. Moreover, it is not known what was the purpose of the work? The purpose of the research should be specified in the Introduction section.
Response 1: We are thankful for pointing out this weakness. The title was changed in “Innovative antimicrobial chitosan/ ZnO/ Ag NPs /citronella essential oil nanocomposite - potential coating for fruits"
Row 94-98: The objective of this research was to obtain a biodegradable antimicrobial film that can be used as packaging material for extending shelf life of fresh fruits. Therefore, chitosan was chosen as base biopolymer, with ZnO and Ag nanoparticles as filler / antimicrobial agents for the fabrication of nanocomposite films. The nanoparticles were loaded with citronella essential oil in order to enhance the antimicrobial activity
We started the assessment of antimicrobial activity through a screening (qualitative evaluation) on several microbial strains, and we are continuing antimicrobial tests by qualitative and quantitative methods on a much larger number of strains specific to food, on both materials, as such and put in contact with fruits (grapes and strawberries). Unfortunately, the whole study is too big to be integrated in just one article. According to the literature these formulations are suitable as coating for fruits.
Point 2: Line 68-73: Please expand on this paragraph. It's too laconic. Please present the conclusions of the research of other scientists.
Response 2: We are thankful for this observation. The paragraph was expanded and comparison with results reported in literature were added (rows 83-103).
Point 3: Line 89-90: Provide reference to table 1 for better clarity (readability) of the text.
Response 3: We have added the reference to table 1 in the first phrase.
Point 4: Table 1: What did Authors suggest when selecting levels of ZnO and Ag NPs in the CZA1-CZA5 sample?
Response 4: We are thankful for the opportunity to enhance our manuscript. The series CZA1, CZA2, CZA3 has the same amount of AgNPs and CEO, but increasing levels of ZnO nanoparticles. The series CZA4, CZA2, CZA5 has the same amount of ZnO nanoparticles and CEO, but increasing concentrations of AgNPs. Therefore, we aimed to obtain two series in which to modify ZnO or AgNPs quantities.
Explanation were added at the end of 2.3 section, with relevant citations:
The concentration of ZnO and AgNPs for the synthesis of CZA films was specially chosen to obtain two series with variable amounts of ZnO or AgNPs. The use of the mixture of two nanoparticles and a natural antimicrobial agent is chosen based on the approaches reported in the literature, especially the synergic and potentiating action. Therefore, in the series CZA1, CZA2 and CZA3 the quantity of AgNPs remains constant, while the ZnO NPs concentration increased. For the series CZA4, CZA2 and CZA5 the amount of ZnO NPs remained constant and the concentration of AgNPs increased. Same quantity of CEO (1 mL) was added to each film.
Point 5: Line 234-236: Please provide a reference to the literature for this fact.
Response 5: We are thankful for this observation. References (31; 33; 45-47) for presence of SPR at ~430nm as function of NPs size and capping agents were added in 3.3.1 section and discussion were enriched.
Point 6: Line 333: ‘chitosan – ZnO NPs – CEO’ it should be ‘chitosan – ZnO – CEO’
Figure 16- 18: Please remove the title from the figure and provide the unit of inhibition diameter.
Section 3.7 (Antimicrobial activity) and section 5 (Conclusions): Please use small letter for names: coli, aureus, albicans and enter a ‘space’ between the value and the unit of inhibition diameter.
Title of section 3 should be “Results and discussion’
Correct the number of section ‘Conclusions’ should be 4 not 5.
Response 6: We are thankful for pointing out these mistakes. The required corrections were made. Inhibition diameter units were added to the reworked figures.
Point 7: Line 405-407: In some countries there have been or are restrictions on the use of PVP in food, hence a limited number of studies may arise.
Response 7: Relevant literature references were added in this section (59 and 60).
Point 8: Line 416-417: And what about the comparison results inhibition diameter of CZ1-CZ4 samples to the value of CEO? It was also present in the CZ1-CZ4 samples, not only ZnO.
Response 8: We have added the CEO inhibition diameters data also to figure 16. We added discussion for CEO influence in CZ films section, rows 496-503 We wish to thank to the esteem reviewer for the opportunity to enrich the paper by comparing our research with relevant results from literature. Detailed discussions and comparison with literature data were also added in section 3.7., including extending shelf life results on grapes and peaches obtained with chitosan – ZnO or chitosan – AgNPs films.
Round 2
Reviewer 3 Report
Authors presented data expressed as means ± standard deviation. The significant differences between the results must be analyzed by analysis of variance (ANOVA). For example, authors should use different lowercase letters, which represent significant differences in the Table 6 and Figures 16-18. Please add point 2.6. Statistical analysis in Materials and Methods section (2).
Author Response
Thank you very much for reviewing our manuscript. We greatly appreciate your thoughtful comments and suggestions that helped improve the manuscript. We have carried out the suggested modifications and revised the manuscript accordingly. We hope you find the revised manuscript acceptable for publication. Thank you once again for your effort.
Point 1: Authors presented data expressed as means ± standard deviation. The significant differences between the results must be analyzed by analysis of variance (ANOVA). For example, authors should use different lowercase letters, which represent significant differences in the Table 6 and Figures 16-18. Please add point 2.6. Statistical analysis in Materials and Methods section (2)
Response 1: We are really thankful for the time spent to help us improve the manuscript.
We have added section 2.6. Statistical analysis
“The data from the antimicrobial assay was analyzed using the analysis of variance (ANOVA) with the help of Microsoft Excel 2016, with XLSTAT 2020.5.1 add-on. Normal distribution of the groups was checked with Shapiro-Wilk test, homoscedasticity of the residuals was assessed by Levene’s test and Tukey’s (HSD) test was used to compare the results and reveal the pairs of films that differed with statistical significance, where p < 0.05.”
The figures 16-18 were replaced with new ones, with letters indicating statistically significant differences between films (p < 0.05). Also, letters indicating statistically significant differences were inserted in table 6.